# Durability of LDPE/UHMWPE Composites under Accelerated Degradation

**DOI:** 10.3390/polym12061241

**Published:** 2020-05-29

**Authors:** Traian Zaharescu, Maria Râpă, Ignazio Blanco, Tunde Borbath, Istvan Borbath

**Affiliations:** 1National Institute for Electrical Engineering (INCDIE, ICPE–CA), Radiochemistry Center; 030138 Bucharest, Romania; 2ROSEAL SA, Odorheiu Secuiesc, 535600 Harghita, Romania; borbath.tunde@gmail.com (T.B.); borbathistvan@roseal.eu (I.B.); 3Department of Materials Processing and Ecometallurgy, “Polytechnica” University, 060042 Bucharest, Romania; rapa_m2020@yahoo.com; 4Department of Civil Engineering and Architecture and UdR-Catania Consorzio INSTM, University of Catania, Viale Andrea Doria 6, 95125 Catania, Italy; iblanco@unict.it

**Keywords:** UHMWPE, LDPE, blends, oxidative degradation, chemiluminescence, DSC

## Abstract

This study presents a detailed analysis of thermal and radiation resistances of low density polyethylene (LDPE)/ultra-high molecular weight polyethylene (UHMWPE) blends containing hydroxyapatite as functional filler and rosemary acting as antioxidant against oxidative degradation. Three main procedures, chemiluminescence (CL), Fourier transform infrared spectroscopy (FTIR), and differential scanning calorimetry (DSC), were applied for the determination of the degree of degradation when these materials are subjected to heat and radiation action. The crystallinity was also assessed for the characterization of diffusion peculiarities. The contributions of the mixing components are discussed based on their oxidation strength. The activation energies required for the oxidative degradation of the studied formulations were calculated.

## 1. Introduction

The resistance of polymer materials against oxidation is one of the main features that define the product durability. The great interest paid to the degradation of polyolefins, the polymers with the largest application areas, explains the deep attention on the investigation of their ageing strengths, which depends strongly on the molecular configuration [1,2]. The addition of protectors like antioxidants [3], active fillers [4,5], and component blending [6,7,8] enlarges the material lifetime, because the mixing components allow the appropriate interactions between the phases and delay of oxidation reactions. The material performances related to the ageing progress intrinsically concern certain functional features related to the molecular structure [9]. Actually, the presence of certain functions in the structure of an additive determines a significant improvement in the oxidation strength [10].

The nanohybrids [11,12,13,14] and modified polymer compositions [15,16,17,18] have opened a new route through which the opposition against degradation preserves the engineering characteristics. In the field of polyolefin blends, the attractive applications in hazardous conditions are turned onto an aggressive environment [19], long-term resistant commodities items [20], cable manufacture [21], medical joints [22], automotive production [23], corrosion protection [24], and ampoules for material irradiation [25]. The main problem for all of the compounds is the characterization of oxidation strength, which determines the material durability. The structural features indicate the perceiving manner by which macromolecules are fragmented during their early stages of degradation [26]. Either in any sort of polyethylene or in ultra-high molecular weight polyethylene (UHMWPE), the free radicals that are formed by bond scissions exist preferentially in the amorphous phase, with a higher abundance where advanced substituted carbon atoms are in initial structures [27]. The progress in the oxidative degradation of various polyethylene structures is sustained by the formation of carbon centered radicals and unsaturation [28]. The development of oxidation grade depends intrinsically on the initial crystallinity degree, which limits the movement of intermediates; the material density, which restricts partially the diffusion of molecular oxygen; the branching level, which defines the amount of high substituted carbon atoms; and the presence of oxidation protector, which delays the propagation chain. The formation of peroxyl intermediates for the initiation of self-catalytic degradation is also related to the conversion of oxidized fragments into alkoxy structures by hydrogen abstraction [29].

The understanding of the oxidation mechanism allows perceiving the lifetime prediction [30] using the Arrhenius equation for the calculation of activation energy (*E*_a_) required for the oxidative degradation. In spite of some restrictions imposed by the ideal degradation scheme [31], this method provides a reliable evaluation of energetic conditions depicting the level of material resistance. In fact, the propagation rate of degradation is determined by the evolution of hydroperoxide amount, which is controlled by the formation and decay rates [32]. Because the blending components present unlike degradation rates and the dissemination of hydroperoxides depends on the material structures and composition, the accumulation of oxidation products that illustrates the formulation resistance would be placed between the component strengths. The thermal resistance of studied low density polyethylene (LDPE)-based compositions defined by the values of activation energy required for their oxidative degradation may characterize the warranted long-term usage, where the external stressors have high intensities for long durations.

The radical mechanism of polymer oxidation [33] explains the formation of oxidation products. However, their concentrations are modified by the presence of certain fillers or protectors. The final purpose of the thermal stability increasing is achieved by the scavenging free radicals in available free volume of structure, like polyhedral oligomeric silsesquioxanes POSS [10], the attraction of intermediates by electronic deficit [34], and the blocking action of antioxidants [35]. These delaying effects lead to a significant increase of material durability by the breaking propagation chain of oxidation. During simulated experiments of oxidative degradation, the considerable contribution of the diffusion of oxygen to the polymer ageing points out the failure conditions when the investigated materials are subjected to accelerated degradation [36]. Because the warranty periods must be extended for a lot of applications, where long-term stability is mandatory, the effects of receipt components have to be revealed. The engineering polymers, like various grades of polyethylene, become the reference materials, when the durability of plastic items is the basic feature for operation security, for example, in nuclear energetics [37]. Owing to their satisfactory oxidation performance and the low degradation rates, especially in the presence of used additives, our materials gain their application ranges, where the service conditions cause early degradation.

One of the several ways that were followed for the improvement of polyethylene product durability is the manufacture of polymer blends with a relevant tendency for high-performance characteristics. The polyethylene blends offer interesting opportunities for their service in various economical areas because they possess appropriate features for various purposes: environmental resistance [38], biological conditions [39], mechanical charges of fabrics [40], recycling [41], high performance thermal insulators [42], and biodegradable composts [43].

The deterioration of the functional properties of polymer blends takes place by weak bond splitting. The degradation mechanisms consisting of the three main stages, initiation, propagation, and termination, involve undiscriminating ways, except the local accumulation of oxidation initiators. Their spreading accompanied by the diffusion of oxygen are decisive factors, which support the increase in the oxygenated stable products of degradation [9,44]. The degradation profile is the result of the overlapping behavior of components and the interpenetration of movable intermediates.

In this study, whose purpose is the evaluation of the most convenient stability of ultra-high molecular weight polyethylene/low density polyethylene formulations, the influence of added components is analyzed. The contributions of rosemary, an antioxidant component, and hydroxyapatite, having the role of radical scavenger, were estimated by three complementary procedures: chemiluminescence (CL), differential scanning calorimetry (DSC), and infrared spectroscopy (FTIR).

## 2. Materials and Methods

### 2.1. Materials

Ultra-high molecular weight polyethylene (UHMWPE) was produced by ARPECHIM Piteşti, Romania, as A31S/130 grade. Its main characteristics are as follows: density of 0.935 g cm^−3^, average numerical molecular weight of 500,000 D, melting temperature of 141.6 °C, and crystallinity degree of 61% (DSC, 10 °C/min). Low density polyethylene (LDPE) type FC 243-55 TIPOLEN was provided by Orbit Polymers SRL, Bucharest, Romania. It is characterized by melting temperature of 114.4 °C and crystallinity degree of 23.9% (DSC, 10 °C/min), flow mass rate (190 °C/2.16 kg) of 2 g/10 min (ISO 1133/B), density (23 °C) of 0.922 g/cm^3^ (ISO 1183-2), tensile strength (MD/TD) of 23/16 MPa (ISO 527-3), and tensile strain at break (MD/TD) 220%/550% of 220%/550% (ISO 527-3). The product is suitable for food contact, being approved by Food Contact Regulations. Rosemary (RM) powder was prepared as in-house product at the INCDIE ICPE CA (Bucharest, Romania) by solvent extraction from rosemary leaves. The raw powder was purified by precipitation with ethanol from aqueous solution. Powder of hydroxyapatite (HAp) nanoparticles was prepared by the procedure reported earlier [45].

### 2.2. Experimental Methods

#### Preparation of Samples

The composition of samples and their labeling are shown in Table 1.

The LDPE/UHMWPE composite (80/20 ratio (wt%)) was melted in a Brabender Plastograph, under a mixing temperature of 180 °C for 10 min and screws rotation rate of 40 rpm. Samples containing 5 wt% HAp and 10 wt% HAp with respect to LDPE and LDPE/UHMWPE blend, respectively, were obtained in the same processing conditions as the control sample. Then, 0.5 wt% RM was added into LDPE/HAp and LDPE/UHMWPE/HAp composites as antioxidant agent.

Specimens for the functional groups and thermal characterization were prepared by pressing of the melted samples in a laboratory press. Several sheets and films with length of 100 mm; width of 100 mm; and thickness of 2 mm and 0.1 mm, respectively, were obtained at the following pressing parameters: temperature of 165 °C, pressing time of 6/3 min (for sheets/films), pressure of 125/150 atm (for sheets/films), and cooling time of 30 min. The plate aspects are presented in Figure 1.

### 2.3. Investigation Methods

#### 2.3.1. Melt Processing

Melt processing behavior of LDPE composites was evaluated by means of torque (at 10 min), melt viscosity (*η*), and power consumption (*P*) [46] obtained from torque-time Brabender diagrams.

#### 2.3.2. FTIR

FTIR spectra were recorded at room temperature using a JASCO 4200A FTIR (JASCO, Tokyo, Japan) spectrometer on neat and heated films having thicknesses of 100 μm over the whole scanning range (400–4000 cm^−1^). The spectra were obtained after 48 scans at a resolution of 4 nm. The sample films destined to spectroscopic investigations were permanently placed in cardboard frames to ensure the exposure of the same point to the light transmission. The thermal degradation of films was accomplished in an electrical oven under isothermal heating at 100 °C for 3 h. For the calculation of carbonyl (CI) and hydroxyl (HI) indices, the internal standard at 2020 cm^−1^ characterizing the CH_2_ stretching was selected, while the accumulations of the main oxygenated products were depicted by the bands at 1720 cm^−1^ for carbonyl groups and 3350 cm^−1^ for the hydroxyl bonds.

#### 2.3.3. Chemiluminescence (CL)

Thermal stability measurements by chemiluminescence were achieved by means of LUMIPOL 3 (Slovak Academy, Bratislava, Slovakia) under isothermal (160 °C, 170 °C, and 180 °C) or nonisothermal (heating rate: 10 and 15 °C min^−1^) regimes, the most suitable condition for ensuring the optimal diffusion of oxygen. The error of temperature measurement was ±0.5 °C. The sliced solid samples with the thickness of 200 μm were placed in aluminum trays because this metal does not spoil information by its oxidation. The emission values were expressed with respect to sample mass, because this form allows the reliable comparison of measurement results. The activation energies from isothermal measurements were calculated using the Arrhenius relationship, which allows the characterization of thermal stability of the studied composite. The evaluation of durability was based on the same representation, where the extrapolation to room temperature was the criterion of comparison.

#### 2.3.4. Differential Scanning Calorimetry (DSC)

Thermal analysis of LDPE composites was performed using a DSC 823e calorimeter (Mettler Toledo, Greifensee, Switzerland). About 10 mg of sample, taken from each sheet weighted with an AS 220/X balance (Radwag, Radom, Poland), were placed into alumina crucibles and examined from room temperature to 200 °C at a heating rate of 10 °C min^−1^. The melting temperature (*T*_m_), melting enthalpy (∆*H*_m_), and degree of crystallinity (*X*_c_) were investigated for all composites, from the first heating scan, with the aid of the software STARe 9.10 from Mettler Toledo, Greifensee, Switzerland. The degree of crystallinity of the LDPE from composites was determined according to Equation (1):(1)$$X_{c}=\frac{\Delta H_{m}}{\Delta H_{100\%,\mathrm{LDPE}}}\times 100\%$$
where Δ*H*_m_ is the measured enthalpy of the melted blends (J/g), and Δ*H*_100%,LDPE_ is the theoretical heat of fusion for a fully crystalline LDPE (290 J/g) [47].

## 3. Results and Discussion

The improvement of polymer resistance against oxidation can be accomplished in different manners, the easiest of which is the addition of protecting compounds [48]. A large application range is covered by phenolic antioxidants whose natural extracts gain more and more areas in ecological products [26,49,50]. The mechanism of proton replacement explains the protective action of the polyphenol-like stabilizers during oxidative ageing of polymers. The inorganic filler, hydroxyapatite, promotes stabilization of polymer materials [45,51,52] and delays oxidation by the adsorption of free radicals on the particle surface, where they are blocked against oxidation. The most important peculiarity of this protection way is depicted by the strength of the joint, determined by the activation energies required for accelerated oxidative degradation.

### 3.1. Melt Processing Behavior

The melt processing behavior of the obtained composites was evaluated from the torque–time curves, and was expressed as torque at 10 min, melt viscosity (*ɳ*), and power consumption (*P*).

Table 2 shows that, by introduction of HAP and RM to the LDPE matrix, the final torque recorded at 10 min was 24 Nxm. As the UHMWPE resin was incorporated into the LDPE matrix, an increase in torque was obtained (36 and 49 Nxm, respectively) compared with those of composites without UHMWPE. The difficulty in the fabrication of the standard polymer processing equipment is already known, owing to its high melt viscosity and lack of fluidity [53]. It was observed (Table 2) that, after 10 min of melt processing, RM agent incorporated into LDPE and UHMWPE did not induce differences in torque compared with composites without RM. The recorded torque for LDPE/UHMWPE, LDPE/UHMWPE/HAp, and LDPE/UHMWPE/HAp/RM composites led to the increase in melt viscosity owing to the UHMWPE added to the LDPE matrix. However, the presence of HAp and RM in composites led to a general decrease of melt viscosity. Accordingly, the melt processing of LDPE composites with UHMWPE, HAp, and RM required lower power consumption (*P*) than that of the control—Table 2. The recorded processing parameters highlight the effect of HAp and RM to improve the flow and melt processing of the LDPE/UHMWPE composite.

### 3.2. FTIR

The FTIR spectra recorded on all compositions revealed the severe changes in the polymer matrix. The oxidation initiated by the formation of peroxyl radicals proceeded by a self-catalytic process and the decay of unsaturation appeared by disproportionation [54,55]. The predominant feature of this oxidation degradation was the significantly higher accumulation rates of carbonyl moieties. The transmission of the 3350 cm^−1^ band increases slowly, while the widths become somewhat larger (Figure 2). The heat treatment applied on the studied samples caused rather the increase in the transmission values of carbonyl bands, which defines the preferable decay manner of peroxyl intermediates generated by the reactions during the propagation step of oxidation [3,56]. The most important tendency noticed from these spectral overlappings was the contribution of filler and additive, which promoted an evident stabilization against the progress of degradation. The carbonyl peak does not present different shoulders, which demonstrates the preference of oxygen reactions on the inner places of the structure than on the chain ends.

The contributions of blending components would be noticed only during the degradation because of the structural similarity and homogenous mixing. The applied heating is a stressing procedure by which the beginning of the oxidation is supported by molecular fragmentation according to the history of manufacture.

The accumulation of oxidation products can be monitored by the modification of spectral transmissions characterizing the tendency of molecular scissions. The propagation of oxidation described by Bolland and Gee’s mechanism, which provides an appropriate approach of degradation results, states the formation of reactive intermediates. The reactions of free radicals generated by the scission of more reactive spots (in our cases, the bonds of multiple substituted carbon atoms), an unsaturation placed preferentially in crystalline zones, and the additives like hydroxyapatite and rosemary active antioxidant components (acting as oxidation inhibitors) produce stabile oxygenated structures, mainly carbonyl and hydroxyl compounds. The routes through which they are formed were previously reported [56]. Owing to the high proportion of LDPE (80 wt%) in the prepared blends, the reported shoulders [57] in the carbonyl vibration do not appear. Their accumulations described by the increase in the carbonyl and hydroxyl indices indicate the oxidation degree of the material. The values of these parameters are illustrated in Figure 3.

The modification of oxidation indices points out the contribution of additives to the improvement of thermal stability. It can be easy noticed that the presence of hydroxyapatite, which superficially promotes adsorption of free radicals, and its increasing loading lead to an improved material able to exhibit a larger durability. If rosemary, the antioxidant component that blocks the oxidation of free radicals by their scavenging, accompanies HAp, the increase in thermal stability is more pronounced. These features suggest the inactivation of reactive moieties (tertiary carbons and double bonds) delaying the deterioration of material functional characteristics, and depict the availability of additives for the simultaneous concurrence of formation and decomposition of further hydroperoxides. The contribution of a couple consisting of HAp and rosemary powder is essential for the production of high-tech polymer materials.

### 3.3. Chemiluminescence (CL)

The evolution of degradation including the differences existing between the sample contents is satisfactorily depicted by isothermal chemiluminescence (CL) spectroscopy (Figure 4). The studied LDPE/UHMWPE compositions conspicuously revealed the presence of two components, if they were manufactured starting from the same monomer. The unmodified blend (LDPE/UHMWPE sample) can be a proper example of the differences in the oxidation rates of components. It is an obvious behavior that both components were simultaneously oxidized, but the curve inclination denoted the preponderance of one of them. The first region, where the curves present a higher degradation rate, belonged to the LDPE. It contained several branching points whose higher substituted carbon atoms became weaker points for the chain scission. In spite of the continuous degradation of UHMWPE over the whole duration of oxidation, its implication was quite visible on the second stage of degradation. This difference disappeared when hydroxyapatite was present in the compositions of LDPE/HAp/RM and LDPE/UHMWPE/HAp/RM samples (Figure 4). This scavenging effect of HAp was previously noticed in the oxidation of poly(lactic acid) [45]. In the comparison of isothermal curves recorded for modified LDPE/UHMWPE blends, the higher stability of compositions containing HAp and rosemary powder can be observed. Regarding the LDPE/UHMWPE/HAp samples, they presented lower oxidation durations or longer induction oxidation times. The degradation kinetics of LDPE/UHMWPE/HAp/rosemary samples proved the significant improvement of oxidation strength owing to the simultaneous stabilization actions of the two added components. On the other hand, the samples containing higher loading (10 wt%) of HAp presented an improved resistance against oxidation compared with the samples with 5 wt% of HAp. This means that the higher content of this inorganic phase offers a satisfactory durability of the products and a detrimental contribution of peroxyl decay to the propagation of material ageing. The concern of stabilization activities of both of the additives is related to the limitation of the activities with respect to free radicals during the earlier stages of degradation, as well as the lack of selectivity regarding the slowing down of oxidation in the cases of these two sorts of polyethylene.

The CL spectra (Figure 4) allowed the evaluation of thermo-oxidation stability with respect to the degradation temperature and material composition. The kinetic analysis of the thermal strength of the studied LDPE/UHMWPE compositions allows achieving an appropriate characterization by the calculation of activation energy values (Table 3).

The higher energies presented by the action of the couple of additives explain that the main source of oxidation was the radicals originating in the chain scissions [56]. The calculated *E*_a_ values were affected by the contribution of LDPE [57,58] and the macroradicals from UHMWPE [59,60].

These energy values are in good agreement with several reported figures [57,61,62,63]. The main process on which the calculation of activation energy was based concerns the conversion of hydroperoxides into final products according to the radical mechanism of the oxidation of polyolefins. According to the mechanism of chemiluminescence measurements, the peroxyl radicals, the source of hydroperoxides, are involved, as described by the following reaction.



The performances of UHMWPE/HAp were previously analyzed [64], but the energetic conditions of degradation were not revealed. The calculated values of *E*_a_ for the thermal oxidation of LDPE/UHMWPE correspond to the beginning and propagation of the degradation process, which involves generation and decay reaction of hydroperoxyls [65]. These activation energy values would be determined by the long-life radicals appearing from UHMWPE [66]. The comparison between the values listed in Table 3 and the similar values reported for polypropylene reveals the higher resistance of present blends. The activation energies of 107–155 kJ mol^−1^ evaluated at high temperatures and 36–49 kJ mol^−1^ calculated for the oxidation occurring at a low temperature were reported [67]. Our values exceed those obtained in the severe condition of ageing by the heating of polypropylene, but they are closer to the value offered for crosslinked polyethylene, XLPE, being placed between 144 and 218 kJ mol^−1^ [68]. This thermal strength of the present LDPE/UHMWPE blends indicates their possible applications for the extended periods not only in medical purposes, but also in the manufacture of pipes, various outer thermal insulations of buildings, automotive items, and various cases for food handling subjected to large temperature variations. Their durability is based on the low oxidation rates, as can be noticed from Figure 4 in the cases of CL curves recorded at a lower inspecting temperature (160 °C).

LDPE/HAp blend presented similar behavior to UHMWPE/HAp under accelerated degradation [69]. The constant gel content in the γ-irradiation of LDPE/HAp compound between 50 and 150 kGy demonstrated the protection effect of filler against the reactions of highly abundant free radicals with oxygen during and post-irradiation. Another analysis regarding the contribution of hydroxyapatite in the energetic exposure of LDPE has pointed out the increase in the crosslinking density of γ-irradiated LDPE [70] from 5 × 10^−4^ mol cm^−2^ at 125 kGy up to 1.1 × 10^−3^ mol cm^−2^ for lower HAp loading (5 wt%) and 1.4 × 10^−3^ mol cm^−2^ for higher HAp loading (15 wt%) at 225 kGy. It was also demonstrated that HAp nanoparticles significantly improved the thermal resistance of polymer support, like poly(lactic acid) [45]. The creation of an interface between hydroxyapatite nanoparticles and the polymer phase, stabilized with rosemary powder, represents a proper way by which long-life radicals are protected against their reaction with oxygen.

The characterization of thermal behavior by increasing temperature in non-isothermal chemiluminescence determinations provides the stability profiles over a large range of temperature. On the low and medium temperature domains, any difference between the emission intensities of oxidizing polymer blends cannot be noticed (Figure 5).

The increase in temperature causes the unlike behavior of compositions. The formulations including hydroxyapatite and rosemary exhibited an extended range of stabilization. The sample containing HAp 5 wt% and rosemary 0.5 wt% began to be oxidized (onset oxidation temperature, OOT) at 239 °C, while the oxidation of the sample including HAp 10 wt% and rosemary 0.5 wt% presented the OOT value at 235 °C (Figure 5a). This aspect completes the degradation picture by a useful proof of improved sample consistency. The other two samples prepared with rosemary presented the thermal profile of up to 225 °C, similar to the reference (unmodified LDPE/UHMWPE). The clearer grouping of thermal answers to the faster heating is illustrated in Figure 5b. The faster the heating, the higher the OOT values. The oxygen feeding, which maintained the progress of oxidation, did not ensure the required amounts for fast reactions during the propagation stage of the thermal ageing. For the individual antioxidant activity of rosemary, the highest concentrations of oxidation initiators were too high at temperatures around 250 °C.

The chemiluminescence determinations provide accurate information on the oxidation state in any moment of degradation. The results can be easily converted into comments on mechanistic aspects, which characterize the intimate evolution of structural modification in polymer components. The studied polyethylene-based nanohybrids present much evidence [70,71,72,73] for their long-term stability, where the filler plays the role of oxidation inhibitor. Their general common feature is the filler blocking action on primary degradation intermediates by the capturing on particle surface.

### 3.4. DSC Measurement

DSC curves for the LDPE composites were collected as the first heating scan (Figure 6).

The thermal parameter values evaluated from the DSC curves of the studied samples are detailed in Table 4.

The presence of two separated endothermic melting peaks (~112 and ~132 °C) was found for the LDPE/UHMWPE composites. These may be ascribed to the mixing of the branched thermoplastic polymer (LDPE) with the super molecular polymer (UHMWPE), indicating the phase separation of the blend before crystallization [74]. Two or even three separated melting peaks were previously reported for LDPE/UHMWPE blends [75,76]. Although the LDPE/UHMWPE blend is miscible in the melt [68], the DSC study revealed a liquid solid phase separation owing to the different rates of crystallization between LDPE and UHMWPE.

According to data shown in Table 4, the melting temperature of UHMWPE and LDPE from composites was lower than those of neat resins (141.6 °C for UHMWPE and 114.4 °C for LDPE), indicating the crystallization independently of the UHMWPE and LDPE chains. A good dispersion of UHMWPE, HAp, and RM into the amorphous phase of LDPE was observed for all blends’ matrices, inducing both a lower melting temperature (from 114.4 °C in the case of neat LDPE to 112.1 °C in the case of LDPE/UHMWPE/HAp/RM composite) and the disruption of the lattice structure of the LDPE matrix, evidenced by the active centers of crystallization diminution. The decrease in crystallinity of LDPE from 23.9% for neat LDPE to 14% for the LDPE/UHMWPE/HAp/RM composite can be also explained by the high aspect ratio of hydroxyapatite and the easy flow in melt processing of HAp and RM (Table 2). Other authors also reported the decrease in the crystallization process owing to the higher interaction of HAp nanoparticles with the amorphous component of the PLA matrix [45] and the incorporation of rosemary into PLA [77].

## 4. Conclusions

The LDPE/UHMWPE formulations available for several applications in hazardous conditions present an improved resistance against oxidation owing to the protective activities of hydroxyapatite and rosemary. The kinetic approach of their degradation provides a complex image of the material answers during oxidative ageing. All of the results confirm the best results with the composition including both of the additives, where the progress in the material oxidation occurs at a slower rate. The individual contributions of additives are based on their availability for the protection of free radicals against the attack of oxygen, as well as the effect of ultra-high molecular weight polyethylene on the decrease of melting enthalpy of the main bending component. The monotone increase of the activation energy required by the oxidative degradation confirms the efficient contributions of UHMWPE, on one hand, and rosemary and hydroxyapatite, on the other hand, can be considered as a proof for the stability improvement of dominant LDPE. Accordingly, the decrease in the oxidation rate values is the intrinsic effect of minor components by the limitation of the free radical amount (in the case of UHMWPE) or restriction of oxidation reactions (in the cases of RM and HAp). Practically, the progress of oxidation concerns the formation of lower amounts of degradation products, which involves the long-term resistance of these hybrids. The most important conclusion is the suitable selection of these studied compositions as a proper choice for the manufacture of various items capable to operate under ageing-promoting conditions.

## Figures and Tables

**Figure 1 polymers-12-01241-f001:**
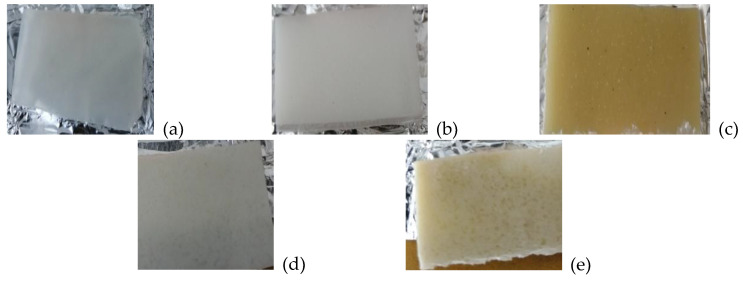
Sheet images for low density polyethylene (LDPE) composites: (**a**) LDPE/ultra-high molecular weight polyethylene (UHMWPE); (**b**) LDPE/hydroxyapatite (HAp); (**c**) LDPE/HAp/rosemary (RM); (**d**) LDPE/UHMWPE/HAp; (**e**) LDPE/UHMWPE/ HAp/RM.

**Figure 2 polymers-12-01241-f002:**
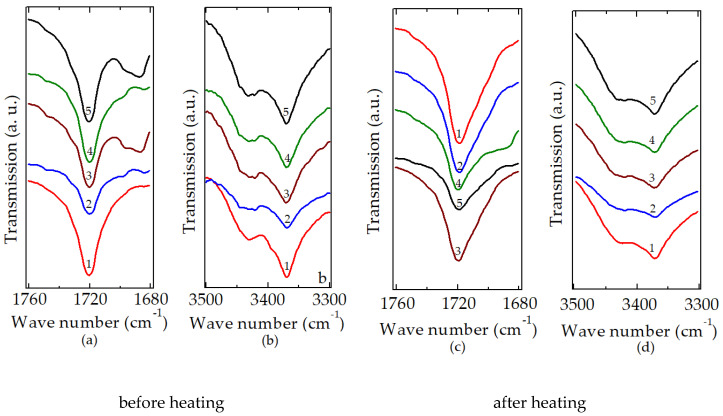
The evolution of carbonyl (**a**,**c**) and hydroxyl (**b**,**d**) bands in LDPE/UHMWPE-based formulations. Indices denote the following: (1) LDPE/UHMWPE, (2) LDPE/HAp, (3) LDPE/HAp/RM, (4) LDPE/UHMWPE/HAp, (5) LDPE/UHMWPE/HAp/RM.

**Figure 3 polymers-12-01241-f003:**
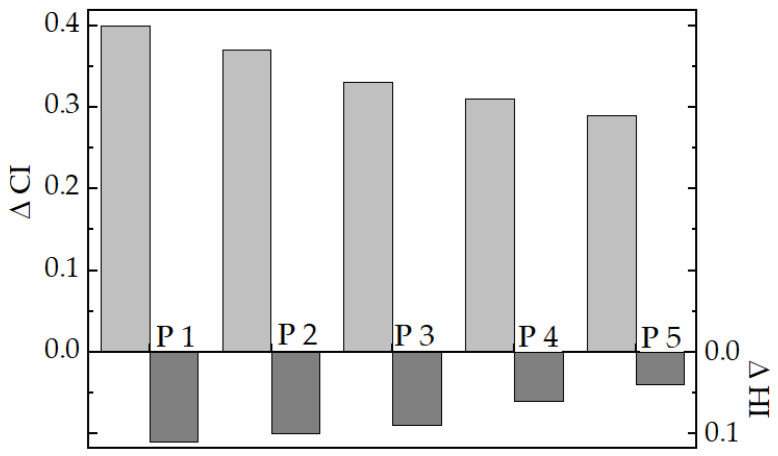
The oxidation indices after the thermal oxidation of studied LDPE/UHMWPE samples. P1 = LDPE/UHMWPE, P2 = LDPE/HAp, P3 = LDPE/HAp/RM, P4 = LDPE/UHMWPE/HAp, P5 = LDPE/UHMWPE/HAp/RM. CI, carbonyl; HI, hydroxyl.

**Figure 4 polymers-12-01241-f004:**
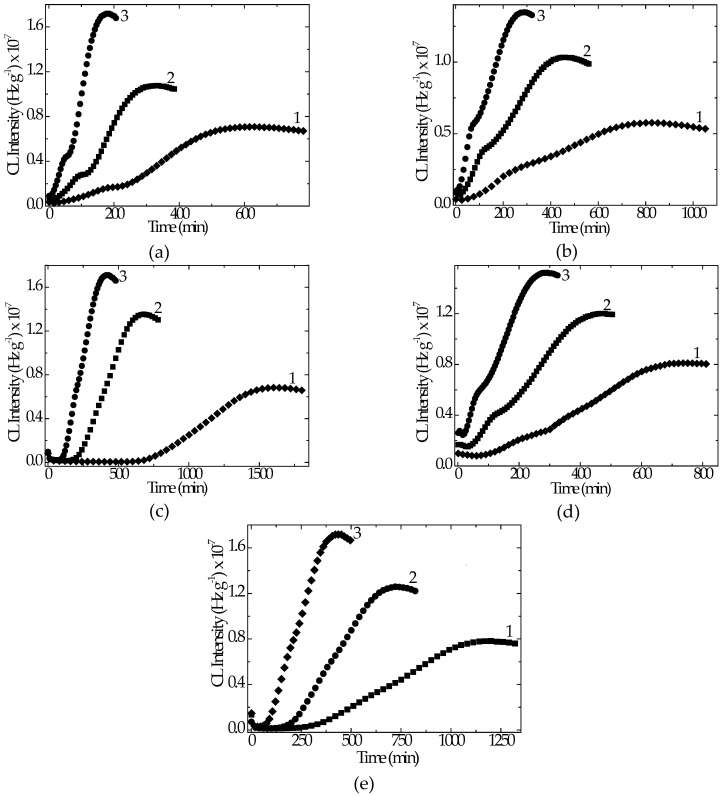
Isothermal chemiluminescence (CL) spectra record on studied samples. Testing temperatures: (1) 160 °C; (2) 170 °C; (3) 180 °C. (**a**) LDPE/UHMWPE, (**b**) LDPE/HAp, (**c**) LDPE/HaP/RM, (**d**) LDPE/UHMWPE/HaP, (**e**) LDPE/UHMWPE/HaP/RM.

**Figure 5 polymers-12-01241-f005:**
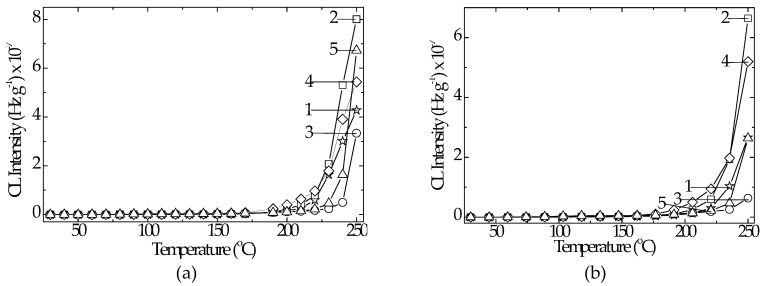
Non-isothermal CL spectra record on the thermally unprocessed LDPE/UHMWPE samples. Heating rates: (**a**) 10 °C min^−1^; (**b**) 15 °C min^−1^. Indices denote the following: (1) LDPE/UHMWPE, (2) LDPE/HAp, (3) LDPE/HAp/RM, (4) LDPE/UHMWPE/HAp, (5) LDPE/UHMWPE/HAp/RM.

**Figure 6 polymers-12-01241-f006:**
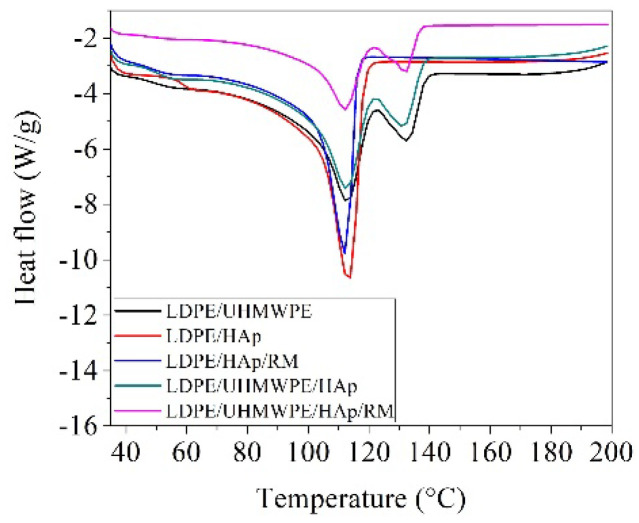
Differential scanning calorimetry (DSC) curves for LDPE composites.

**Table 1 polymers-12-01241-t001:** Composition of samples. LDPE, low density polyethylene; UHMWPE, ultra-high molecular weight polyethylene; HAp, hydroxyapatite; RM, rosemary.

Sample	LDPE, wt%	UHMWPE, wt%	HAp, wt%	RM, wt%
LDPE/UHMWPE	80	20		
LDPE/HAp	95		5	
LDPE/HAp/RM	94.525		4.975	0.5
LDPE/UHMWPE/HAp	72	18	10	
LDPE/UHMWPE/HAp/RM	71.64	17.91	9.95	0.5

**Table 2 polymers-12-01241-t002:** Melt processing parameters for LDPE composites evaluated from torque–time Brabender diagrams.

Sample	Torque [Nxm]	*η* [Nxm/rpm]	*P* [kW]
LDPE/UHMWPE	49	1.225	205.14
LDPE/HAp	24	0.6	100.48
LDPE/HAp/RM	24	0.6	100.48
LDPE/UHMWPE/HAp	36	0.9	150.72
LDPE/UHMWPE/HAp/RM	36	0.9	150.72

**Table 3 polymers-12-01241-t003:** Calculated activation energies from isothermal chemiluminescence determinations.

Sample	Oxidation Rate [au g^−1^ sec^−1^] × 10^3^			Relationship	Correlation Factor	E_a_ [kJ mol^−1^]
	160 °C	170 °C	180 °C			
LDPE/UHMWPE	0.238	1.001	1.902	y = 52.77 − 20.42 x	0.97847	169.7
LDPE/HAp	0.222	1.008	1.901	y = 54.27 − 20.10 x	0.97622	175.4
LDPE/HAp/RM	0.119	0.534	1.094	y = 55.29 − 21.82 x	0.98232	181.4
LDPE/UHMWPE/HAp	0.141	0.458	1.435	y = 57.40 − 22.71 x	0.99999	188.8
LDPE/UHMWPE/HAp/RM	0.136	0.474	1.512	y = 59.32 − 23.55 x	0.99998	195.7

**Table 4 polymers-12-01241-t004:** Melting temperature (*T*_m_), enthalpy of melting (Δ*H*_m_), crystallization temperature (*T*_c_), and degree of crystallinity (*X*_c_) for LDPE composites.

Sample	Δ*H*_m_,_LDPE_ [J/g]	*T*_m_,_LDPE_ [°C]	Δ*H*_m_,_UHMWPE_ [J/g]	*T*_m_,_UHMWPE_ [°C]	*X*_c_,_LDPE_ [%]	*X*_c_,_UHMWPE_ [%]
LDPE/UHMWPE	46.7	112.8	12.2	132.7	16.1	4.2
LDPE/HAp	83.8	113.2			28.8	
LDPE/HAp/RM	85.8	112.1			29.5	
LDPE/UHMWPE/HAp	41.4	112.3	11.5	131.8	14.2	3.9
LDPE/UHMWPE/HAp/RM	40.8	112.1	14.5	132.0	14.0	5.0

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
