# Peer review of "Durability of LDPE/UHMWPE Composites under Accelerated Degradation"

_polymers, 2020, doi:10.3390/polym12061241_

Round 1

Reviewer 1 Report

This manuscript presented an analysis of thermal and radiation resistances of  a compound of low density polyethylene (LDPE)/ultra-high molecular weight polyethylene (UHMWPE) blends containing hydroxyapatite as functional filler and rosemary acting as antioxidant against oxidative degradation. Different characterization techniques were used. The contributions of the mixing components are well discussed based on their oxidation degree. 

It is recommended to add some mechanical tests such as micro-indentation, impact and tensile test, in the case that they are available and if possible. These tests would help the main results to become more obvious and more understandable.

Thank you for your attention.

Author Response

Dear reviewer,

The authors of this manuscript offer their thanks for your involvement in the editorial way. We carefully analyzed your comments thinking on the improvement of this paper.

As you correctly remarked, we present an academic study on the thermal resistance of LDPE/UHMWPE blend in the presence of two additives acting as stabilization component. We should be glad to tell you that the second part (under preparation) of this study concerns the radiation effect on these present compositions. Basically, we started this detailed investigation on the characterization of stability features conducted for the evaluation of durability and exploration of optimal solution for the improvement of oxidation resistance.

You are right suggesting the achievement of mechanical testing for the depicting the behavior of materials over operation period. However, there are two main barriers, may be restrictions: first circumstance is determined by available time, which is not enough long for reliable determinations for all samples having in mind that a serious investigation has to be done on neat and thermally aged materials. Second aspect is related to the availability of experimental infrastructure. Unfortunately, these measurements would be planned for any other next studies.

So, we bag your pardon that we cannot follow your pertinent recommendation.

Finally, the authors are believing that these present answers correspond to the reviewers’ consensus and the approval for acceptance will be obtained for the authors’ gladness.

In behalf of all authors,

Traian Zaharescu

Reviewer 2 Report

1.     For the study of material oxidation resistance, it is recommended to conduct a long-term thermal oxygen aging according to the actual service conditions, and then compare the performance.
2.     The application of the copolymer prepared in this paper was not illustrate,and the difference between the copolymer and LDPE or UHMWPE was not explain.
3.     In the study, there is not only a lack of theoretical research on oxidation process, but also no engineering application guidance significance. It is suggested to conduct detailed research and analysis before publication.

Author Response

Dear reviewer,

The authors of this manuscript offer their thanks for your involvement in the editorial way. We carefully analyzed your comments thinking on the improvement of this paper.

  1. For the study of material oxidation resistance, it is recommended to conduct a long-term thermal oxygen ageing according to the actual service conditions, and then compare the performance.

The durability of materials is an essential feature that recommends them for long-term service. As any reader examines the Table 3, the studied materials have the high values of activation energy calculated for their thermal degradation. For comparison, we added the similar values for other engineering polymer, polypropylene, provided by a prodigious team.

Added text

The comparison between the values listed in Table 3 and the similar values reported for polypropylene reveals the higher resistance of present blends. The activation energies of 107-155 kJ mol-1 evaluated at high temperatures and 36-49 kJ mol-1 calculated for the oxidation occurred al low temperature were reported [67]. Our values exceed those obtained in the severe condition of ageing by the heating of polypropylene, but they are more closed to the value offered for crosslinked polyethylene, XLPE, being placed between 144 and 218 kJ mol-1 [68]. This thermal strength of the present LDPE/UHMWPE blends indicates their possible applications for the extended periods not only in medical purposed, but also in the manufacture of pipes, various outer thermal insulations of buildings, automotive items, the cases for food handling. Their durability is based on the low oxidation rates as it can be noticed from Figure 4 in the cases of CL curves recorded at lower inspecting temperature (160 °C).

  1. The application of the copolymer prepared in this paper was not illustrated and the difference between the copolymer and LDPE or UHMWPE was not explained.

The component mixture LDPE/UHMWPE was separately analyzed in the former edition of manuscript. Either LDPE or UHMWPE were studied in detail because of their large spectra of applications. The studies on the degradation of LDPE are largely known. Concerning the degradation mechanism in UHMWPE, several exceptional papers were published:

  1. Jacobson, Oxidation ultra high molecular weight polyethylene (UHMWPE). Part 1: Interpretation of the chemiluminescence curve recorded during thermal oxidation. Polym. Degrad. Stab. 2006, 91, 2126-2132.
  2. Bracco, E.M. Brach del Prever, M. Cannas, M. P. Luda, L. Costa, Oxidation behaviour in prosthesis UHMWPE components sterilised with high energy radiation in a low-oxygen environment. Polym. Degrad. Stab. 2006, 91, 2030-2038.
  3. K., Wannomae, S. D. Christensen, A. A. Freiberg, S. Bhattacharyya, W. H. Harris, & O. K. Muratoglu. The effect of real-time aging on the oxidation and wear of highly cross-linked UHMWPE acetabular liners. Biomater., 2006, 27(9), 1980–1987.
  4. Oral, K. K. Wannomae, S. L. Rowell, O. K. Muratoglu, Migration stability of a-tocopherol in irradiated UHMWPE, Biomater., 2006, 27, 2434-2439.
  5. Dalbong, K. Jacobson, S. Jonsson, Methods for determining the spatial distribution of oxidation in ultra high molecular-weight polyethylene prosthesis. Polym. Degrad. Stab. 2007, 92, 437-447.
  6. Shen, L. Costa, Y. Xu, Y. Cong, Y. Cheng, X. Liu, J. Fu, Stabilization of highly crosslinked ultra high molecular weight polyethylene with natural polyphenols. Polym. Degrad. Stab. 2014, 105, 197-205.

The both mixed polymers were intensively studied, so we should consider that separate approach of these materials charges too much the manuscript. Please, adopt our point of view.

  1. In this study, there is not only a lack of theoretical research on oxidation process, but also no engineering application significance. It is suggested to conduct details research and analysis before publication.

Your strong point of view is correct especially for a review manuscript. We agree it, because it represents a large approach of any scientific subject. Taking into account that this manuscript is an original work, that the previous literature has reported a lot of results describing the effects of thermal degradation or including ageing mechanisms, that the interpretation of results has a large part of academic contributions, that the applications are always based on experimental information being abundant in this manuscript, that the correlation between the investigation and our data curation conducts toward a clear interpretation of oxidation resistance including the evaluation of activation energies, we believe that the aim of this manuscript is professionally attained. We may compare this manuscript with the other last papers published by us in peer journals:

  1. Zaharescu, T. Borbath, V. Marinescu, A. M. Luchian, I. Borbath. Improvement of thermal stability of EPDM by radiation crosslinking for space applications. J. Therm. Anal. Calorim., 138, 2445-2455 (2019). Doi: 10.1007/s10973-019-08581-2.
  2. Zaharescu, A. R. Caramitu, V. Marinescu. Stability analysis of PA6/ethylene elastomer blends for severe ageing applications. Polym. Bull., 77, 565-583 (2020). Doi: 10.1007/s00289-19-02761-8.
  3. Zaharescu, I. Blanco, F. A. Bottino. Surface Antioxidant Activity of Modified Particles in POSS/EPDM Hybrids. Appl. Surf. Sci., 509, 144702 (2020). Doi:10.1016/j. apsusc.2019.144702.
  4. M. Lupu, T. Zaharescu, E. M. Lungulescu, M. Râpă, H. Iovu. Availability of PLA/SIS blends for packaging and medical applications. Radiat. Phys. Chem., 172, 108696 (2020). DOI: 10.1016/j.radphyschem.2020.108696
  5. Zaharescu, C. Tardei, V. Marinescu, M. Râpă, M. Iordoc. Interphase surface effects on the thermal stability of hydroxyapatite/poly(lactic acid) hybrids. Ceramics Int., 46, 7288-7297 (2020). Doi: 10.1016/j.ceramint.2019.11.223
  6. Râpă, L. M. Stefan, T. Zaharescu, A. M. Suciu, A. A. Turcanu, E. Matei, A. M. Predescu, I. Antonic, C. Predescu. In vitro biocompatibility and stabilization effect of AgNPs on some plascticized PLA/collagen bionanocomposites. Appl. Sci., 10, 2265 (2020). Doi: 10.3390/app10072265

All of them were prepared in the same manner and with the same professional concern. Let us hope that you will trust in our proper scientific contribution.

Finally, the authors are believing that these present answers correspond to the reviewers’ consensus and the approval for acceptance will be obtained for the authors’ gladness.

In behalf of all authors,

Traian Zaharescu

Reviewer 3 Report

Zaharescu et al. prepared a series of composites and studied their degradation mechanisms. The manuscript seems interesting. Before, making final decision some questions need to be discussed:

  1. Please provide the synthesis details of polyethylene.
  2. The decrease in molecular weight supports the degradation mechanism. Could GPC be a better choice? Please comment on it. Need more experimental evidence to prove the resistance against oxidation.
  3. It is not clearly discussed what is the role of hydroxyapatite and rosemary.
  4. In general, the whole manuscript should be concise and to the point. Some sections need to be rewritten.  

Author Response

Dear reviewer,

The authors of this manuscript offer their thanks for your involvement in the editorial way. We carefully analyzed your comments thinking on the improvement of this paper.

  1. Please, provide the synthesis details of polyethylene.

Please, receive our apologizing for the misunderstanding this question. It is a general practice that the manufacturer does not report his production technologies. We used materials, LDPE (type FC 243-55 TIPOLEN from Orbit Polymers SRL, Romania) and UHMWPE (A31S/130 grade from ARPECHIM Piteşti, Romania) as received products, so, we are not able to provide any technical information on production procedure.

  1. The decrease in the molecular weight supports the degradation mechanism. Could GPC be a better choice? Please comment on it. Need more experimental evidence to prove the resistance against oxidation.

During the oxidative degradation several simultaneous factors acting on the ageing material determine an overall behavior. The fragmentation of polymer macromolecules generates reactive intermediated, whose sizes depends on the number of weak sites along the macromolecules. Finally, the thermal stability depends on the concentrations of these entities, being reflected in the oxidation rates and activation energies involved in the progress of degradation. The GPC investigation provide essential information on the scission manner, but the final effect of degradation can be evaluated by a chemical procedure like chemiluminescence. Beside the lack of this useful instrument, we intend to present the material general behavior under the thermal oxidation, not structural study.

  1. It is not clear what is the role of hydroxyapatite and rosemary.

We previously mentioned the role of these additional compounds. However, in this new edition of manuscript, we specially describe the role of hydroxyapatite and rosemary extract for a clear emphasizing of their contribution.

Added text

The improvement of polymer resistance against oxidation can be accomplished by different ways, through which the easiest manner is the addition of protecting compounds [48]. A large application range is covered by phenolic antioxidants whose natural extracts gain more and more areas in ecological products [26,49,50]. The mechanism of proton replacement explains the protective action of the polyphenol-like stabilizers during oxidative ageing of polymers. The inorganic filler, hydroxyapatite promotes stabilization of polymer materials [45,51,52] delays oxidation by the adsorption of free radicals on the particle surface, where they are blocked against oxidation. The most important peculiarity of this protection way is depicted by the strength of joint determined by the activation energies required for accelerated oxidative degradation.

[48] Seguchi, T.; Tamura, K.; Shimada, A.; Sugimoto, M.; Kudoh, H.  Mechanism of antioxidant interaction on polymer oxidation by thermal and radiation ageing. Radiat. Phys. Chem. 2012, 81, 1747-1751.

[49] Olejnik, O.; Masek, A.; Kiersnowski, A. Thermal analysis of aliphatic polyester blends with natural antioxidants. Polymers, 2020, 12, 74.

[50] Kirschweng, B.; Tátraaliai, D.; Földes, E.; Pukánszky, B. Natural antioxidants as stabilizers for polymers. Polym. Degrad. Stab. 2017, 145, 25-40.

[51] Tzankova-Dintcheva, N.; Infurna, G.; Baiamonte M.; D’Anna, F. Natural compounds as sustainable additives for biopolymers. Polymers, 2020, 12, 732.

[52] Tsai, S-W.; Yu, W-X.; Hwang, P-A.; Hsu, Y-W.; Hsu, F-Y. Fabrication and characteristics of PCL membranes containing strontium-substituted hydroxyapatite nanofibers for guided bone regeneration. Polymers, 2019, 11, 1761.

  1. In general, the whole manuscript should be concise and to the point. Some sections need to be rewritten.

According to this last suggestion, we reread the manuscript and corrected that we believe to be the interpretable words or phrases.

Finally, the authors are believing that these present answers correspond to the reviewers’ consensus and the approval for acceptance will be obtained for the authors’ gladness.

In behalf of all authors,

Traian Zaharescu

Round 2

Reviewer 2 Report

The significance of the material application and performance improvement should be detailed in the introduction.

Figure 5 is not clear, so it is recommended to use logarithmic coordinates.

Reviewer 3 Report

The manuscript is ready to be accepted. 

Author Response

Dear reviewer,

Thank you for your precious suggestion.